# Integrated and DC-powered superconducting microcomb

Chen-Guang Wang[1,2,3], Wuyue Xu[1,2,3], Chong Li[1,2,3], Lili Shi[1], Junliang Jiang[1], Tingting Guo[1], Wen-Cheng Yue[1,2,3], Tianyu Li[1,2,3], Ping Zhang[1], Yang-Yang Lyu[1,2], Jiazheng Pan[2], Xiuhao Deng[4,5], Ying Dong [6], Xuecou Tu [1,5], Sining Dong [1,3], Chunhai Cao[1], Labao Zhang [1,5], Xiaoqing Jia [1,5], Guozhu Sun[1,5], Lin Kang[1,5], Jian Chen [1,2], Yong-Lei Wang [1,2,3] ✉, Huabing Wang [1,2] ✉ & Peiheng Wu [1,2] ✉

Frequency combs, specialized laser sources emitting multiple equidistant frequency lines, have revolutionized science and technology with unprecedented precision and versatility. Recently, integrated frequency combs are emerging as scalable solutions for on-chip photonics. Here, we demonstrate a fully integrated superconducting microcomb that is easy to manufacture, simple to operate, and consumes ultra-low power. Our turnkey apparatus comprises a basic nonlinear superconducting device, a Josephson junction, directly coupled to a superconducting microstrip resonator. We showcase coherent comb generation through self-started mode-locking. Therefore, comb emission is initiated solely by activating a DC bias source, with power consumption as low as tens of picowatts. The resulting comb spectrum resides in the microwave domain and spans multiple octaves. The linewidths of all comb lines can be narrowed down to 1 Hz through a unique coherent injection-locking technique. Our work represents a critical step towards fully integrated microwave photonics and offers the potential for integrated quantum processors.

Frequency combs serve as high-precision rulers for frequency and time measurement, playing a pivotal role in a wide variety of modern science and technologies[1–5], including optical clocks, LIDAR, spectroscopy, arbitrary waveform generation, and optical neural networks. Over the past two decades, integrated combs have garnered significant research interests[5–18], leading to miniaturized and chip-based photonic systems[7]. However, most on-chip frequency combs, such as integrated semiconductor mode-locked lasers and microresonator-based Kerr combs, mainly operate in the optical frequency domain. A fully integrated frequency comb functioning in the microwave domain remains elusive, impeding the advancement of chip-based microwave

spectroscopy and integrated quantum circuits, which typically require precise microwave control. Here, we address these challenges by introducing an all-superconductor-based microcomb, featuring an elegantly simple structure, effortless operation, and ultra-low power consumption.

Our superconducting frequency comb is fully integrated, comprising two fundamental superconductor devices: a Josephson junction directly coupled to a superconducting coplanar waveguide (CPW) resonator, as illustrated in Fig. 1a (see Supplementary Fig. 1 for a photo of our device). Device fabrication is achieved through straightforward procedures utilizing standard photolithography and electron beam

[1]Research Institute of Superconductor Electronics, School of Electronic Science and Engineering, Nanjing University, Nanjing, China. [2]Purple Mountain Laboratories, Nanjing, China. [3]National Key Laboratory of Spintronics, Nanjing University, Suzhou, China. [4]Shenzhen Institute for Quantum Science and Engineering, Southern University of Science and Technology, Shenzhen, China. [5]Hefei National Laboratory, Hefei, China. [6]College of Metrology Measurement and Instrument, China Jiliang University, Hangzhou, China. ✉e-mail: yongleiwang@nju.edu.cn; hbwang@nju.edu.cn; phwu@nju.edu.cn

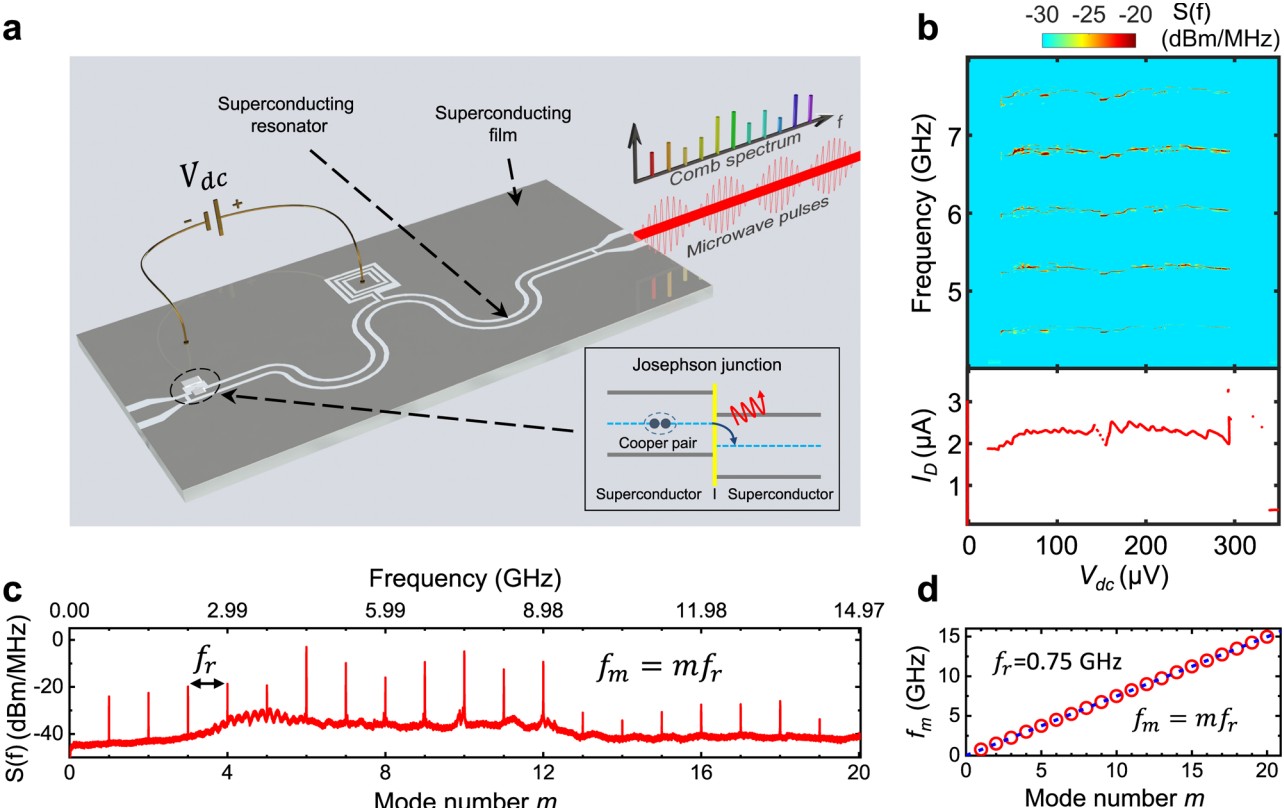

**Fig. 1 | Integrated and DC-powered superconducting microcomb. a** Illustration of the device. Frequency comb emission is generated when a DC voltage-biased ($V_{dc}$) Josephson junction is coupled to a superconducting CPW resonator. **b** The power spectral density (top) and flowing supercurrent through the Josephson junction (bottom) as a function of DC bias voltage. **c** Frequency comb spectrum obtained at $V_{dc} = 43\,\mu V$. **d** Extracted emission frequencies (open dots) from (**c**). The dashed line represents fitting to the comb formula $f_m = m f_r$.

evaporation techniques (See Methods). The Josephson junction consists of two superconductors connected by a weak link or a thin insulating barrier. It acts as an ideal voltage-to-frequency converter, emitting photons at the Josephson frequency $f_J = 2eV_{dc}/h$, where $2e$ is the charge of a Cooper pair, $V_{dc}$ is the DC voltage bias across the junction, and $h$ is Planck's constant. Meanwhile, the superconducting CPW resonator is another crucial superconductor device widely used in high-sensitive detectors and circuits for qubit control and read-out[19]. We demonstrate that the combination of these two fundamental superconducting elements can generate a coherent frequency comb (Fig. 1b, c). Its on-chip generation of pulse waves and inherent compatibility with production, operation, and integration into circuit quantum electrodynamics[20–22] offers the potential for miniaturized, low-cost, and energy-efficient quantum processors.

The Josephson junction-coupled superconducting resonator has drawn significant research interest due to its capability not only to operate as ultrasensitive sensors, such as for single photon detection[23] and/or thermometry[24], but also to function as a coherent photon source[25–34]. Particularly, in the strong coupling regime, the device generates stable microwave lasing[25] and demonstrates remarkably low noise[26]. However, in previous investigations, the coupled devices were all operated in single mode, resulting in continuous-wave emissions. Their properties under a multimode operation have not been explored.

## Results

### Coherent comb generation

We demonstrate that a Josephson junction coupled to a superconducting resonator can generate a self-phase-locked coherent

frequency comb under multimode operation. The coupling between a Josephson junction and a resonator is proportional to the Josephson energy $E_J h I_c/2e$, where $I_c$ is superconducting critical current[25,28]. To directly ensure strong coupling, we employ a Josephson junction with a large $I_c$. This is achieved by fabricating a sizable Josephson junction (approximately 4 μm²), resulting in a relatively large supercurrent of $I_D \approx 2\,\mu A$ in the lasing state (Fig. 1b), which surpasses by more than two orders of magnitude compared to that in the continuous-wave Josephson laser[25]. To achieve multimode operation within the frequency range of interest (typically 1–10 GHz), we fabricate a superconducting resonator with a low free spectrum range of 0.75 GHz. The strong coupling allows stable microwave lasing when a dc voltage bias $V_{dc}$ **20** μV is applied to the Josephson junction (Fig. 1b).

Figure 1c shows the comb spectrum measured at $V_{dc} = 43\,\mu V$. First, we will prove that the observed spectrum in Fig. 1c indeed represents a coherent frequency comb. The raised background between 3–10 GHz in Fig. 1c is a consequence of the limited bandwidth of amplifiers. Therefore, our subsequent investigations will focus on the frequency range in the detection bandwidth. The frequencies of a comb are given by the formula: $f_m = f_0 + m f_r$, where $f_0$ denotes the carrier offset frequency, $f_r$ represents the repetition frequency, and $m$ stands for the mode number[1–5]. The extracted mode frequencies are shown in Fig. 1d, align perfectly with fitting to the comb formula. The result indicates a negligible offset frequency ($f_0 \approx 0$ GHz). Consequently, our superconducting microcomb can be expressed by a simplified comb formula:

$$f_m = m f_r \tag{1}$$

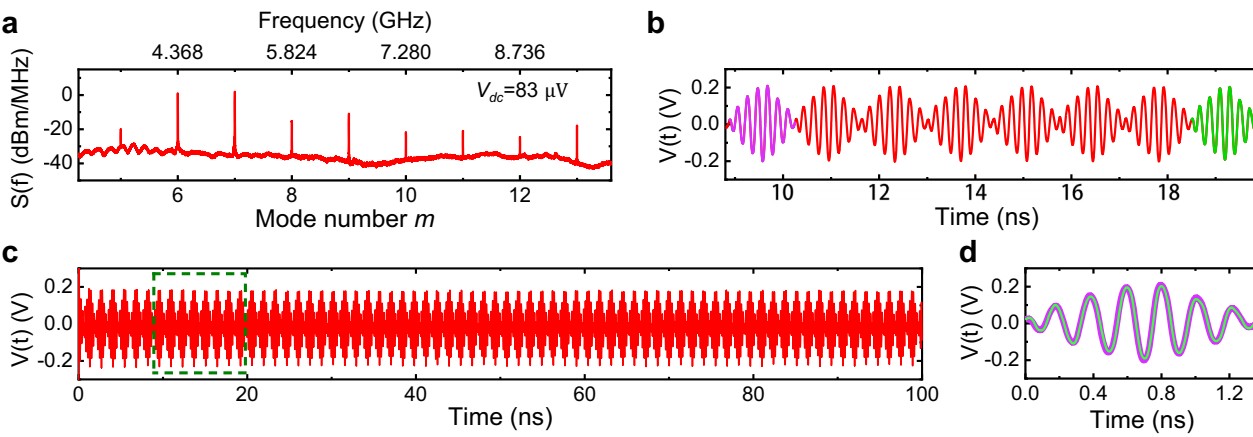

**Fig. 2 | Coherent pulse-wave. a** Comb spectrum obtained at $V_{dc} = 83\,\mu V$. **b–d** Pulse waveforms corresponding to the spectrum in (**a**). The zoom-in waveform (**b**) of the region enclosed by the green-dashed box in (**c**) displays two highlighted pulse waves with purple and green, which overlap perfectly in (**d**).

Additionally, Fig. 1c, d demonstrate that the repetition frequency $f_r$ nicely aligns with the free spectrum range (0.75 GHz) of the half-wave resonator.

The evenly spaced spectral lines alone are not sufficient for correlations among the comb teeth. To examine the coherence state of the emitted signal, we first proved that each individual mode has a stable phase using the heterodyne detection technique (Supplementary Fig. 2). One direct consequence of phase-coherent comb is the generation of pulses in the time domain[1–5]. Figure 2a displays a comb spectrum, and its corresponding time-dependent waveform is shown in Fig. 2b, c. The waveform reveals a sequence of microwave pulses, indicating that all spectrum modes are phase-locked and maintain a stable phase relationship. Moreover, every pulse displays identical features, as seen in Fig. 2d by the perfect overlap of two pulse waves. This indicates that all pulses are in-phase, and also suggests a carrier offset frequency of 0 Hz, thus confirming that all the spectral lines share a uniform phase. The coherence of the comb modes can be further validated through a unique coherent injection-locking effect, as demonstrated below.

The emission power of each individual comb line, ranging up to subpicowatt (see Supplementary Fig. 3), is consistent with that of the continuous-wave (or single-mode) source[25]. The microcomb's performance could be further enhanced by refining the structures and parameters of the Josephson junction-coupled resonators[26]. The stability of the frequency comb is determined by the linewidth of the spectral lines (see Supplementary Fig. 4). In general, the overall linewidth is influenced by the quality factor of the resonator, the stability of the DC bias voltage, and environmental electromagnetic and thermal noise. Moreover, our experiments demonstrate that the linewidth of our comb increases quadratically with the mode number (or frequency), as illustrated in Supplementary Fig. 5. This finding is consistent with the observations in quantum-limited optical combs[35,36].

**Coherent injection-locking effect**

Injection-locking is a widely used technique for effectively narrowing the linewidth of a laser source[25,26,37,38]. In optical frequency combs, it has been demonstrated only in a few cases in quantum cascade lasers by injecting a radio-frequency (RF) modulation at the cavity roundtrip frequency[39–42]. In our superconducting microcomb, we showcase a distinct coherent injection-locking effect that can be achieved by injecting an external microwave at any arbitrary comb mode. For instance, we select an arbitrary mode, such as $m_{inj} = 7$, as the injection tone (Fig. 3a). Then, an injection signal $f_{inj}$ is applied and swept around $f_7 = 5.26$ GHz. We measure the spectra at frequencies $f_{sen}$ around various sensing modes $m_{sen}$ (Fig. 3a). The resulting spectra maps are illustrated in Fig. 3b–d (additional results in Supplementary Fig. 6).

We find that all modes are simultaneously locked within the same injection frequency range $\Delta f_{inj} = 8.08$ MHz (see Fig. 3b, d for definition and Fig. 3k for extracted values). This reaffirms that all spectral lines are coherently phase-locked.

This unique coherent injection-locking effect results in an ultra-high-resolution comb, with the linewidth of all comb teeth significantly narrowed down to ≤1 Hz, as shown in Fig. 3e–g and their insets. It's worth noting that the observed linewidth of 1 Hz is constrained by the resolution bandwidth of the spectrum analyzer used in the experiments[26], implying the potential for even narrower linewidths.

The injection-locking range $\Delta f_{inj}$ of our microcomb widens with the injection power $P_{inj}$ (Supplementary Fig. 7). Specifically, when $m_{sen} = m_{inj}$ (Fig. 3c, f, i), the injection-locking phenomena replicate those observed in the continuous-wave source[25,26], and can therefore be explained by the Adler's theory[43]. However, the results for the comb modes with $m_{sen} \neq m_{inj}$ cannot be described by Adler's equation (see Supplementary Fig. 8a, b). We find that the locked sensing frequency range $\Delta f_{sen}$ (see Fig. 3b–d for definition) is proportional to $m$ (Fig. 3l). This implies that the radiation frequencies of the injection-locked comb satisfy the comb formula (1) over the entire locked frequency range. Consequently, this introduces an in-situ tunable superconducting microcomb, allowing for adjustable $f_r$ within the range of $\Delta f_{inj}/m_{inj}$.

The coherent injection-locking leads to a unique frequency-pulling effect, particularly noticeable when $m_{sen} \neq m_{inj}$. Figure 3h–j illustrate these effects when $f_{inj}$ is fixed at an off-resonance tone (the corresponding on-resonance injection results are shown in Supplementary Fig. 9), shifted from the free-running emission tone by $\Delta f_{off} = 8$ MHz. When $m_{sen} = m_{inj}$, the emission line is gradually pulled toward $f_{inj}$ with increasing injection power $P_{inj}$, ultimately locking at $f_{inj}$ (Fig. 3i). This behavior mirrors that observed in a continuous-wave source[25,26]. However, the remarkable outcome arises when $m_{sen} \neq m_{inj}$. In this case, although no injection tone is applied around each sensing mode, notable emission lines are induced at the corresponding tones, termed "induced-injection tone" and labeled as $f_{id}$ in Fig. 3b, d, h, 3j. The linewidth of the $f_{id}$ lines is considerably narrower than those of the free-running emissions and the side-band harmonics (see Supplementary Fig. 10). Furthermore, neither $f_{id}$ emission nor Kerr comb generation are observed with RF injection under zero DC bias voltage (Supplementary Fig. 11). These indicate that the $f_{id}$ emissions originate from the mutual interactions among Josephson photons, injection photons, and the resonator's multi-modes, setting the $f_{id}$ signals apart from the Kerr combs generated directly from RF pumping[44].

When $m_{sen} \neq m_{inj}$, as depicted in Fig. 3h, j, while the emissions of all the modes are pulled toward $f_{id}$ tones with increasing $P_{inj}$, the $f_{id}$ tones themselves also shift with $P_{inj}$. Consequently, this results in the locked

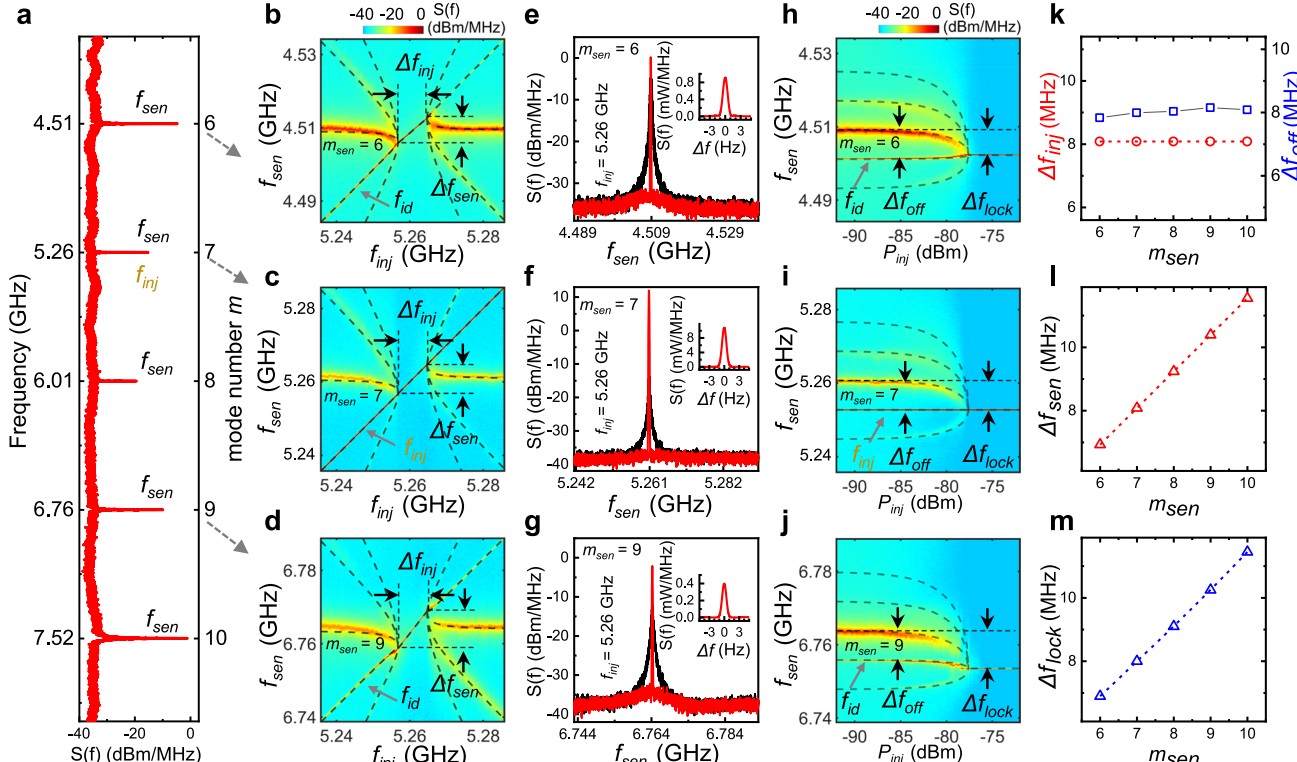

**Fig. 3 | Coherent injection-locking effect. a** A comb spectrum showing five observed modes. **b**–**d** Spectrum maps obtained by sweeping the injection frequency $f_{inj}$ around $f_7$ at a fixed power ($P_{inj} = -82$ dBm), and at various sensing modes $m_{sen} = 6$ (**b**), 7 (**c**), and 9 (**d**), respectively. The emission signals are locked within the range $\Delta f_{inj}$ and $\Delta f_{sen}$. **e**–**g** Comparison of spectra between free-running (black) and injection-locking (red) states. Insets show the linewidth, which is ≤1 Hz for all locked modes. **h**–**j** Spectra maps obtained by varying injection power ($P_{inj}$) with an off-resonance tone ($f_{inj} = f_7 \cdot \Delta f_{off}$; $f_7 = 5.26$ GHz, $\Delta f_{off} = 8$ MHz). The shifts of the locked tone from the free-running tones are indicated by $\Delta f_{lock}$. The long-dashed lines in (**b**–**d**, **h**–**j**) are fittings to extended Alder's Eq. (2). **k**–**m** Plots of $\Delta f_{inj}$ and $\Delta f_{off}$ (**k**), $\Delta f_{sen}$ (**l**), and $\Delta f_{lock}$ (**m**) as functions of mode number $m$.

frequency range $\Delta f_{lock} \neq \Delta f_{off}$ (see Fig. 3h, j). Figure 3m demonstrates that $\Delta f_{lock}$ is also proportional to $m$ with $\Delta f_{lock} = \frac{m}{m_{inj}} \Delta f_{off}$, perfectly aligned with the requirements of the comb formula (1) for the emissions within the locking range.

There is no existing theory that can describe our observed coherent injection-locking effect. We have derived an extended Adler's equation to quantitatively describe these unique phenomena. As analyzed above, in the locking range $\Delta f_{inj}$, all comb emissions are described by the comb formula (1). In the unlocking range outside $\Delta f_{inj}$, the emission signals are given by (please refer to Method for detailed derivation):

$$f_{m,n} = \frac{m}{m_{inj}}(f_{inj} + f_h) + nf_h \quad (2)$$

Here, $f_h$ denotes the repetition frequency of the harmonic emissions induced by off-resonance injection, and $nf_h$ represents the $n$th harmonic shift. The $f_h$ is given by:

$$f_h = (f_{m_{inj}} - f_{inj})\sqrt{1 - \left(\frac{\Delta f_{inj}/2}{f_{m_{inj}} - f_{inj}}\right)} \quad (3)$$

where $\Delta f_{inj} = \alpha \sqrt{P_{inj}}$ (with $\alpha$ as a constant related to cavity losses), and $f_{m_{inj}}$ is the mode frequency at $m_{inj}$. Notably, when $m = m_{inj}$, the extended Adler's Eq. (2) reduces to the standard Adler's equation. The perfect fittings in Fig. 3b–d, 3h–j indicate that our extended Adler's Eq. (2) accurately describes all the coherent injection-locking effects in our superconducting microcomb.

In previous studies of continuous-wave superconducting lasers, the downconversion of higher-order Josephson frequencies to the resonator's fundamental mode has been demonstrated[25]. However, the upconversion of Josephson photons to higher mode has not been reported. In Fig. 4a, we present a comb spectrum (red) within our typical experimental bandwidth of 3–10 GHz. The applied DC bias voltage across the Josephson junction is $V_{dc} = 36.387$ μV, corresponding to a Josephson frequency $f_J = 17.60$ GHz. As illustrated in Fig. 4a, all the comb modes with $f_m < f_J$ are generated through downconversion, and any modes with $f_m > f_J$ (if existing) would result from upconversion. However, emissions with $f_m > 17.60$ GHz lie beyond the bandwidth of our spectrum measurements, preventing their direct observation.

The coherent injection-locking effect provides an advanced methodology for sensing comb emissions beyond the detection bandwidth. To demonstrate this, we perform injection-locking measurements using various injection tones with $m_{inj} = 1$, 6, 54, as illustrated in Fig. 4a, while selecting a fixed sensing tone within the detection bandwidth, such as $m_{sen} = 7$. Figures 4b–d display the coherent injection-locking effects. The injection-locking phenomenon arises from the nonlinear interaction between emission tone and external injection signal[43]. Since no injection signal is applied at $m = 7$, the coherence injection-locking effect displayed in Fig. 4d suggests the presence of a comb emission tone at $f_{54} = 39.28$ GHz, which is significantly higher than the Josephson frequency $f_J = 17.6$ GHz. This observation provides evidence for the upconversion process of Josephson photons to higher modes. The presence of comb emissions at $m = 1$ (Fig. 4b) and 54 (Fig. 4d) indicates that our Josephson

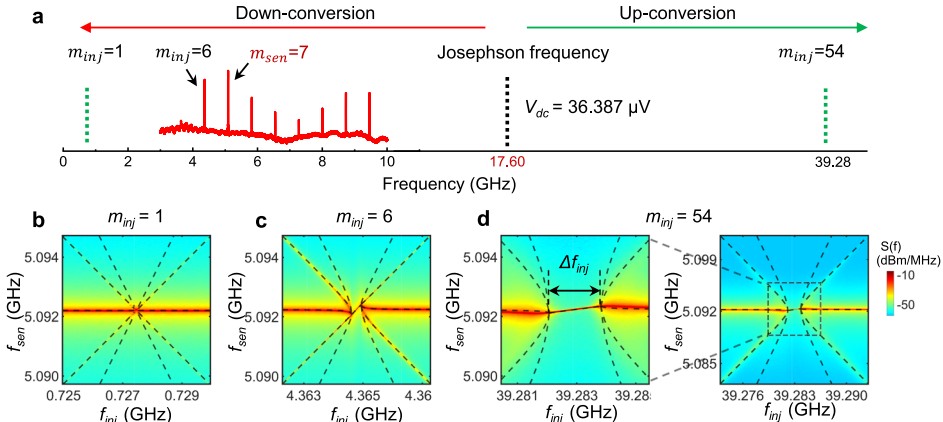

**Fig. 4 | Up- and down-conversion of Josephson photons into multiple modes.**
**a** Illustration of the up- and down-conversion. A comb spectrum (red) is obtained at $V_{dc} = 36.39\,\mu V$, corresponding to a Josephson frequency of 17.6 GHz. **b**–**d** Coherent injection-locking effects with $m_{sen} = 7$ and for injections at $m_{inj} = 1$ (**b**), 6 (**c**), and 54 (**d**), respectively. The dashed lines are fittings to extend Alder's Eq. (2).

microcomb spans a frequency range exceeding five octaves, a span typically challenging to achieve in semiconductor-based frequency combs. It is essential to note that this octave value is limited by the upper-frequency limit of the microwave generator used in our experiments. Hence, it is highly likely that even higher modes could exist beyond this range.

## Discussion

We have demonstrated a fully integrable superconducting microcomb. Unlike previously reported superconductor-based frequency combs[44–52], which all operated under microwave pumping using externally supplied, expensive, and energy-consuming microwave sources, our superconducting microcomb is driven solely by a DC bias voltage. This makes it highly desirable for scalable and on-chip integration. The initiation of our microcomb requires simply tuning on a low-power DC bias source, with a minimal input power as low as 40 pW (Fig. 1b)−approximately nine orders of magnitude lower than the semiconductor-based frequency combs. This presents significant advancement for ultrasensitive and energy-efficient applications.

The exceptional coherence of our comb enables a unique coherent injection-locking effect previously inaccessible in frequency combs, leading to an ultra-high-resolution and tunable comb. This innovation introduces unique functionalities, such as the generation of coherent subcombs through four-wave mixing, as showcased in Supplementary Fig. 12. The simultaneous up- and down-conversion of Josephson photons significantly expands the comb's frequency range over multiple octaves. These capabilities offer added flexibility and tunability for applications of superconducting microcomb technology.

This DC-biased superconducting microcomb with ultra-low power consumption (down to a few tens of picowatts) can work at ultra-low temperatures, aligning seamlessly with the operating conditions of superconducting quantum circuits. For a typical dilution refrigerator, which has a cooling power of hundreds of microwatts at 20 mK, in principle, it is feasible to integrate up to millions of the combs at the base temperature based on their energy efficiency. Our comb can serve as a multifrequency microwave source for multiplexed quantum measurement[53]. An important potential application of our superconducting microcomb is the development of a low-cost, on-chip arbitrary waveform generator. This could be achieved by controlling the intensities of the comb teeth[54], which is possible by integrating our superconducting comb with a series of frequency-tunable superconducting resonators serving as adjustable filters[55]. This advancement promises substantial benefits for future quantum technologies. Moreover, implementing superconducting frequency combs

in the terahertz domain could be feasible by using intrinsic Josephson junctions in high-temperature superconductors[56]. This approach could also extend the technology's working temperature range.

## Methods
### Device fabrication

We fabricated four superconducting microcombs (#1-#4), and detailed parameters are listed in Supplementary Tables 1. The superconducting resonators are fabricated using a superconducting Nb film for device #1 and a Ta (α-phase) film[57] for devices #2, #3, and #4. The film was sputtered on a 10 mm by 10 mm sapphire substrate (C-plane, thickness 650 μm). Standard photolithography followed by reactive ion etching in a CF4 was then used to define the resonators. The Al/AlO$_x$/Al (Al thicknesses 40 nm/80 nm) Josephson junction was fabricated using standard double-angle evaporation and lift-off techniques. To ensure good electrical contact between the Josephson junction and the resonator layers, the sample was ion-beam milled to remove residual oxides and resist residue from the surface of the resonator before the double-angle evaporation.

### Experiments

The devices are mounted in a dilution refrigerator with a base temperature of 20 mK. A complete circuit is shown in Supplementary Fig. 13. We adopt a similar approach to refs. [25,30,31] for the low noise biasing scheme to measure the tunneling current and provide a stable voltage bias to the device. The bias voltage is supplied by an on-chip voltage divider circuit consisting of a 10 Ω shunt resistor and a 10 Ω reference resistor. The current through the device is then measured via the voltage drop $V_r$ across the 10 Ω reference resistor. The current $I_D$ and voltage $V_{dc}$ of the device is given by $I_D = V_r/10$ and $V_{dc} = 10I_{bias} - 20I_D$. Here, $V_r$ is obtained from a nano voltmeter (Keithley 2182 A) and $I_{bias}$ is the output from a current source (Keithley 6221). Additional filtering for the circuit is provided by two 100 μF chip capacitors. Additionally, all low-frequency lines are heavily filtered outside the PCB with multi-pole RC and PI low-pass filters located in the mixing chamber of the dilution refrigerator.

The device's output signal is amplified by an amplifier chain consisting of a cryogenic amplifier (+42 dB) and two room-temperature amplifiers (+32 dB). The spectra are acquired by an Agilent N9010A spectrum analyzer (S/A), and the waveforms are recorded using a Keysight MSOV334A digital oscilloscope (D/O). The injection signal to the resonator is generated by an Agilent N5183A analog signal generator (S/G) and is attenuated by a low-temperature attenuation chain to assure that the thermal contribution of photons to the cavity is negligible.

## Extended Adler's equation

In the unlocking range outside $\Delta f_{\text{inj}}$, the emission signals for $m_{\text{sen}} = m_{\text{inj}}$, which can be described by Adler's theory[43], are given by:

$$f_{m_{\text{inj}},n} = (f_{\text{inj}} + f_h) + n f_h \qquad (3)$$

where $f_h$ is the repetition frequency of the harmonic emissions. The first term, $f_{\text{inj}} + f_h$, accounts for the comb's emission at $m_{\text{inj}}$, while $n f_h$ represents the $n$th harmonic shift. The $f_h$ is given by:

$$f_h = (f_{m_{\text{inj}},0} - f_{\text{inj}})\sqrt{1 - (\frac{\Delta f_{\text{inj}}/2}{f_{m_{\text{inj}},0} - f_{\text{inj}}})}$$

where $\Delta f_{\text{inj}} = \alpha\sqrt{P_{\text{inj}}}$ (with $\alpha$ as a constant related to cavity losses). The comb's emission at $m_{\text{inj}}$ corresponds to $n = 0$, therefore we obtain $f_{m_{\text{inj}},0} = f_{\text{inj}} + f_h$ from Eq. (3). Utilizing the superconducting microcomb formula (1) (see the main text), we can express the comb's emissions for any arbitrary $m$ as $f_{m,0} = \frac{m}{m_{\text{inj}}} f_{m_{\text{inj}},0} = \frac{m}{m_{\text{inj}}}(f_{\text{inj}} + f_h)$. By adding the $n$th harmonic shift $n f_h$, we derive an extended Adler's Eq. (2), as shown in the main text, for the coherent injection-locking in our superconducting combs.

## Data availability

All the data that support the findings of this study are available on the public repository https://doi.org/10.6084/m9.figshare.25480621

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

## Acknowledgements

This work is supported by the National Key R&D Program of China (2021YFA0718802 (Y.-Y.L., Y.-L.W., and H.W.), 2018YFA0209002 (Y.-L.W.), and 2023YFF0718400 (Y.D.)), the National Natural Science Foundation of China (62274086 (Y.L.W.), 62288101 (H.W.), 12204434(Y.D.), and 62271245 (X.T.)), Postdoctoral Fellowship Program of CPSF (W.-C.Y. and Y.-Y.L.), Jiangsu Outstanding Postdoctoral Program (W.-C.Y. and Y.-Y.L.), Shenzhen Science and Technology Program (KQTD20200820113010023 (X.D.)), and Jiangsu Key Laboratory of Advanced Techniques for Manipulating Electromagnetic Waves.

## Author contributions

Y.L.W., H.W., and P.W. conceived and supervised the project. C.-G.W. designed and fabricated the devices, as well as conducted the experiments. L.S., J.J., T.G., C.C., X.J., and J.C. provided support for the fabrication of resonators. T.L., Y.-Y.L., and G.S. assisted in the fabrication of Josephson junctions. W.X., C.L., W.-C.Y., X.T., and L.K. provided support for optical lithography. W.X., S.D., P.Z., Y.-Y.L., J.P., and L.Z. assisted in microwave spectrum and waveform measurements. C.G.W. and Y.L.W. performed the analysis and interpretation of the experimental data. C.-G.W., X.D., Y.D., and Y.-L.W. conducted the theoretical analysis. C.-G.W. and Y.-L.W. wrote the manuscript. Y.-L.W. and H.W. edited the manuscript.

## Competing interests

The authors declare no competing interests.
