## [Peer Review File · Nature Communications]

REVIEWER COMMENTS

Reviewer #1 (Remarks to the Author):

This paper describes a comprehensive study on the DC generation of microwave frequency combs in a superconducting device. This device integrates a superconducting microwave resonator with a 0.75 GHz free-spectral-range (FSR) and a Josephson junction (JJ) with a relatively large critical current compared to that in a typical superconducting qubit. Due to the nonlinearity of the JJ, the system presents microwave frequency combs whose comb repetition rate is defined by the FSR of the resonator. The novelty of this work is obviously the realization of microwave frequency combs by applying only a DC voltage to the junction. This distinguishes the current work from previous study that utilized a RF source for comb generation. The authors demonstrate coherent nature of the generated combs by studying periodic pulses in time-domain. The authors also thoroughly studied injection locking effect due to an additional coherent microwave tone, which is another interesting point of this paper.

Overall, the paper has its own novelty, obviously, and all the data are well presented. I therefore support the publication of this paper after addressing my comments below.

1. In the abstract and introduction section of this paper, the authors gave a motivation of their work by comparing with optical frequency combs. However, microwave and optical combs are operating in totally different frequency ranges and therefore should have distinct applications. I recommend the authors provide more clear motivation on why microwave frequency combs are unique and interesting to study.
2. The authors repeatedly mentioned miniaturized, low-cost, energy-efficient quantum processors as an application of their superconducting frequency comb. I do not see clear connection of their combs and superconducting quantum information processing. Please discuss this point. .
3. I feel like the authors put quite a lot of efforts in explaining coherent injection locking in their device. The data in Figure 3 show that the introduction of the external injection tone at f_{inj} leads to the narrowing of the comb peaks at several f_{sen} below < 1 Hz. It would be nice to provide the data showing linewidth of comb teeth at several f_{sen} versus the power of the injection tone. Also, how does such narrowing effect depend on the V_{dc} ?
4. Regarding the linewidth of comb peaks generated only by the DC bias, which parameters in their system determines their free-running linewidth? How are the peak linewidths related to the linewidth of their resonant modes?
5. It seems like the linewidth narrowing effect here is likely related to parametric process due to the nonlinearity of the device. Could the authors please discuss the origins of this narrowing effect?
6. Here the authors only focus on the behavior of individual linewidths. What about the stability of the frequency combs with/without the injection tone? The stability means the stability of the spacing (or the repetition frequency) of the combs.
7. The authors also discuss induced-injection tones. I think these tones are also frequency combs

generated by the injection tones. If not, how the induced tones are different from combs generated by a RF tone?

8. The sensing of up-converted combs using a low-frequency sensing tone (Within the detection bandwidth) is also quite intriguing. Could the authors clarify the procedure for this experiment? It seems like the authors send an injection tone at 39.28 GHz and monitored some sensing tone. But to me, it is still unclear how the authors prove the presence of the combs at up-converted frequencies.

9. Also, why did the authors choose the specific value of the DC bias, 36.387 μV ?

10. Again, in the discussion part, the authors claim that their combs are energy efficient as their method does not require RF signals for the comb generation. What could be the cases that such energy efficient generation is advantageous? What is the semiconductor-based frequency comb? Is it optical?

11. In the last paragraph, the authors brought a very interesting application of their comb, which is on-chip arbitrary waveform generation. I totally agree with this point, but would like to know if this method has specific advantages over cryogenic CMOS-based waveform generation.

Reviewer #2 (Remarks to the Author):

This transformational technique of generating frequency combs with a very low current injection and coupling to a superconducting resonator is inspired and high quality work. The addition of the injection locking is very important, but the observation of the coherent subcombs is the most useful in terms of confirming that this is a coherent comb structure.

If there are any negatives, it seems that there are statements about the possibility of developing on-chip AWGs that go too far and may actually distract from the achievements outlined. For example, the implementation of that involves pulse-shaping methods that may or may not be available at cryogenic stages.

While it can easily be calculated what power is in the peaks, it would be useful for the authors to estimate that and put it in the context of quantum circuits, i.e. drives and readouts, so the applicability for these generators at base temperature can be easily translated to that technology.

Overall an excellent idea and well organized and written paper.

Responses to comments and recommendations of Reviewer 1

Reviewer recommendation:

This paper describes a comprehensive study on the DC generation of microwave frequency combs in a superconducting device. This device integrates a superconducting microwave resonator with a 0.75 GHz free-spectral-range (FSR) and a Josephson junction (JJ) with a relatively large critical current compared to that in a typical superconducting qubit. Due to the nonlinearity of the JJ, the system presents microwave frequency combs whose comb repetition rate is defined by the FSR of the resonator. The novelty of this work is obviously the realization of microwave frequency combs by applying only a DC voltage to the junction. This distinguishes the current work from previous study that utilized a RF source for comb generation. The authors demonstrate coherent nature of the generated combs by studying periodic pulses in time-domain. The authors also thoroughly studied injection locking effect due to an additional coherent microwave tone, which is another interesting point of this paper.

Overall, the paper has its own novelty, obviously, and all the data are well presented. I therefore support the publication of this paper after addressing my comments below.

Our response:

We express our gratitude to you for your positive feedback, conditional endorsement of publication, and insightful comments that have significantly enhanced the quality of the paper. In the following sections, we address each of your comments comprehensively.

Comment#1:

1. In the abstract and introduction section of this paper, the authors gave a motivation of their work by comparing with optical frequency combs. However, microwave and optical combs are operating in totally different frequency ranges and therefore should have distinct applications. I recommend the authors provide more clear motivation on why microwave frequency combs are unique and interesting to study.

Our response

Thank you for your thoughtful comments. You accurately noted the differences in application domains between optical and microwave frequency combs, stemming from their distinct operating frequency ranges. For example, in spectroscopy measurements using frequency combs, a comb in the microwave domain could advance electron paramagnetic resonance and ferromagnetic resonance spectroscopy techniques, which have a wide range of applications in chemistry, physics, and biological sciences.

Another unique potential application of our on-chip superconducting frequency combs is to advance quantum technologies. For instance, the control of quantum circuits and/or qubits typically requires the use of microwave pulses, which could be directly supplied by a microwave frequency comb. Of course, a tunable comb with adjustable pulse waves is essential for future developments. The future development of integrated arbitrary waveform generators based on our on-chip microwave frequency comb would significantly benefit quantum computing (please refer to our response to Comment #1 of Reviewer 2 regarding the possible development of arbitrary waveform generators).

We have revised our manuscript to include these clarifications, emphasizing the unique applications of microwave frequency combs in comparison to their optical counterparts in the abstract.

Comment#2:

2. The authors repeatedly mentioned miniaturized, low-cost, energy-efficient quantum processors as an application of their superconducting frequency comb. I do not see clear connection of their combs and superconducting quantum information processing. Please discuss this point.

Our response

We appreciate your inquiry, which complements the concerns raised in comment #1 regarding the potential applications of our frequency comb. The generation of pulse wave signals and the potential for developing on-chip Arbitrary Waveform Generators (AWGs) are directly relevant to the control and readout processes in superconducting quantum circuits and qubits, which typically requires the use of microwave pulse waves. These applications highlight the integral role that our comb technology could play in miniaturizing, cost-reducing, and enhancing the energy efficiency of quantum processors by providing precise, scalable, and energy efficient pulse-waves essential for quantum computing. We have rephrased the statement in the INTRODUCTION section to clarify this point.

Comment#3:

3. I feel like the authors put quite a lot of efforts in explaining coherent injection locking in their device. The data in Figure 3 show that the introduction of the external injection tone at f_{inj} leads to the narrowing of the comb peaks at several f_{sen} below ≈ 1 Hz. It would be nice to provide the data showing linewidth of comb teeth at several f_{sen} versus the power of the injection tone. Also, how does such narrowing effect depend on the V_{dc} ?

Our response

Thank you for your constructive suggestions. In response, we have conducted the corresponding experiments and provided below the data (Fig. S1) detailing the dependence of the linewidth of comb teeth on the injection power across various sensing modes (the injection mode is $m_{inj}=9$), which are included in the Supplementary Information as Supplementary Figure 10. It is worth noting that the linewidth variation is not a direct measure of injection locking. As shown in the below Fig. S1e, the linewidth of $m=9$ drops at much lower injection power than those for the other sensing modes. This is because at low injection power the interaction between comb emissions and injection signal is neglectable, and the linewidth is independently determined by the injection signal when the injection signal intensity is higher than the emission signal, as illustrated by the spectrum line in Fig. S1g for $P_{inj}=-129$ dBm. The consistent evolutions of the background signals (or free-running emission signals) in Figs. S1g-S1h for all modes are consistent with the *coherent* injection locking effect.

Regarding the effect of V_{dc} , as documented in our manuscript, the linewidth of the injection-locked comb is consistently observed at 1 Hz. However, its crucial to note that this observed 1 Hz linewidth is constrained by the frequency resolution of the spectrum analyzer. Consequently, the actual linewidth could potentially be much narrower than 1 Hz. Within the frequency resolution limit of 1 Hz, we did not observe a notable effect of the DC bias voltage (V_{dc}) on the linewidth of the locked comb teeth.

Fig. S1 | Dependence of linewidth on injection power. (a) Spectrum of the free running comb. (b-d), Spectra maps obtained by varying injection power (P_{inj}) under $m_{inj}=9$ for $m_{sen}=7, 9, 11$, respectively. (e) Extracted linewidths plotted as a function of P_{inj} . (f-h) Spectrum lines with selected P_{inj} values.

Comment#4:

4. Regarding the linewidth of comb peaks generated only by the DC bias, which parameters in their system determines their free-running linewidth? How are the peak linewidths related to the linewidth of their resonant modes?

Our response

In general, the overall linewidth of a comb is determined by the quality factor of the resonator, the stability of the DC bias voltage, and environmental electromagnetic and thermal noise. Our recent experiments have shown that the linewidth of our comb

increases quadratically with the mode number (or frequency), as demonstrated in Fig. S2. This behavior is consistent with observations in optical combs, as reported in the literatures [J. Opt. Soc. Am. B 24, 8 (2007); Phys. Rev. Lett. 122, 203902 (2019)]. We have also observed that the linewidth varies with the DC bias voltage, a phenomenon illustrated in Fig. S2 (compare Figs. S2A-S2H under $V_{dc}=245\ \mu\text{V}$ to Figs. S2I-S2P under $V_{dc}=68.5\ \mu\text{V}$). However, we have not find a consistent rule of the linewidth variations with DC bias. More systematic experiments and the underlying mechanisms of this variation are still under investigation. We have included Fig. S2 in the Supplementary Information as Supplementary Figure 11 in the revised manuscript.

Comment#5:

5. It seems like the linewidth narrowing effect here is likely related to parametric process due to the nonlinearity of the device. Could the authors please discuss the origins of this narrowing effect?

Our response

The nonlinearity of the Josephson junction in our device indeed plays a crucial role in the observed linewidth narrowing associated with the injection locking process. This effect can be attributed to a combination of factors including phase synchronization, noise reduction, and mode stabilization, which arise from the nonlinear interaction between the emission tone and the external injection tone. The foundational theory of the injection locking process was introduced by Robert Adler in his paper published in Proc. IRE, vol. 34, pp. 351-357 (1946), cited as Ref. 41. To be noted, Adlers theory primarily addresses the injection locking of *single-mode* oscillators.

In our manuscript, we have developed phenomenological equations to quantify the coherent injection locking process of our frequency comb. This approach is inspired by Adler's equations and is supported by our experimental observations. Although our equations can quantitatively describe the observed coherent injection locking process of our comb, the underlying microscopic mechanisms remain elusive. This aspect falls beyond the scope of our current experimental research efforts, yet it underscores a compelling direction for future theoretical work.

Comment#6:

6. Here the authors only focus on the behavior of individual linewidths. What about the stability of the frequency combs with/without the injection tone? The stability means the stability of the spacing (or the repetition frequency) of the combs.

Our response

We conducted time-dependent measurements for the comb spectra with and without the injection tone. The results are shown Fig. S3. Both free-running (Figs. S3a-S3f) and injection-locked (Figs. S3g-S3l) combs show very stable frequency lines over time. The stabilities of the spacing between emission lines are mainly determined by the linewidth and do not change over time (Figs. S3m and S3n). We have included these data in the Supplementary Information as Supplementary Figure 12 in the revised manuscript.

Fig. S3 | Stability of frequency combs. (a) a comb spectrum. (b-g) Time dependence of the free-running comb spectrum maps. (h-m) Time dependence of the injection-locked comb spectrum maps. (n and o) Time dependence of line spacing for the free-running comb (n) and injection-locked comb (o).

Comment#7:

7. The authors also discuss induced-injection tones. I think these tones are also frequency combs generated by the injection tones. If not, how the induced tones are different from combs generated by a RF tone?

Our response

Thank you for raising this excellent point. The previously reported RF-pumped combs originate from four-wave mixing, akin to optical Kerr frequency combs. These typically require a high nonlinearity of the superconducting resonator fabricated using highly nonlinear superconducting materials, such as niobium-titanium nitride. In our work, the resonator was fabricated using pure niobium (and/or tantalum) film, which has relatively low dynamic inductance and low nonlinearity. In our experiments, we did not observe any comb generations when *only* applying an injection signal into the resonator and *without* applying a DC bias voltage across the Josephson junction, as shown in Fig. S4b below. That is, there were no induced-injection tones observed by the RF injection tone under zero DC bias (no Kerr comb generation from RF injection). Therefore, we conclude that the induced tones are generated from the mutual interactions among Josephson photons (driven by DC bias across the Josephson junction), injection photons, and the resonator's multi-modes, thus setting the induced tones apart from the Kerr combs directly generated by a RF tone. We have included this figure as Supplementary Figure 7 and added the corresponding discussion to clarify this point in the revised manuscript. The detailed microscopic mechanism of the induced tones can be an interesting and important project for future in-depth theoretical studies.

Fig. S4 | Comparison between microwave injection with and without comb emission. (a) Spectrum with injection tone $m_{inj}=9$ (6.578 GHz) and at $V_{dc}=195.57$ μ V (with comb emission). **(b)** Spectrum with the same injection tone and at $V_{dc} = 0$ V (without comb emission).

Comment#8:

8. The sensing of up-converted combs using a low-frequency sensing tone (Within the detection bandwidth) is also quite intriguing. Could the authors clarify the procedure for this experiment? It seems like the authors send an injection tone at 39.28 GHz and monitored some sensing tone. But to me, it is still unclear how the authors prove the presence of the combs at up-converted frequencies.

Our response

Your understanding of our experimental procedure is accurate. We applied a series of injection tones around 39.28 GHz, corresponding to the 54th comb mode, and conducted spectral measurements at a sensing tone within our detection bandwidth. Specifically, the results presented in Fig. 4 were obtained near 5.09 GHz, which aligns with the 7th comb mode. The injection locking phenomenon, as discussed in response to Comment #5, arises from the nonlinear interaction between the emission tone and the injection tone at (or near) identical frequencies [Ref. 41]. Therefore, the observation of coherence injection locking effect in Fig. 4d demonstrates the presence of comb emissions at 39.28 GHz, the 54th comb mode (given that no injection tone applied at the 7th comb mode). Moreover, because the 54th comb tone (39.28 GHz) exceeds the Josephson frequency emitted directly from the Josephson junction under a DC bias of 36.387 μ V. This provides evidence for the up-conversion process in our experiment. To clarify this, we have included the corresponding discussion in the revised manuscript to eliminate any ambiguity.

Comment#9:

9. Also, why did the authors choose the specific value of the DC bias, 36.387 μ V?

Our response

First, the DC bias voltage value is determined and calculated from the circuit (with sampling resistors) shown in Supplementary Figure 9. The selection of the DC bias voltage in Fig. 4 was strategically made to utilize a minimal DC bias that yields a sufficiently low Josephson frequency, facilitating the demonstration of up-conversion. Essentially, any DC bias value that results in a Josephson frequency below the upper frequency limit (<40 GHz) of our microwave generators bandwidth is suitable for this experiment. Specifically, a DC bias of 36.387 μ V was chosen because it corresponds to a Josephson frequency of 17.6 GHz, fitting well within our experimental parameters.

Comment#10:

10. Again, in the discussion part, the authors claim that their combs are energy efficient as their method does not require RF signals for the comb generation. What could be the cases that such energy efficient generation is advantageous? What is the semiconductor-based frequency comb? Is it optical?

Our response

Yes, the semiconductor-based frequency comb operates mainly in the optical frequency domain. Moreover, semiconductor-based combs typically require a large pumping threshold, ranging from tens of milliwatts to watts, which makes them unsuitable for on-chip integration at ultralow temperatures. The ultra-low dissipation of our DC-biased comb, with an on-chip input power of tens of picowatts, is perfectly suited for integration with superconducting quantum circuits at dilution refrigerator temperatures.

For a typical dilution refrigerator, which has a cooling power of hundreds of microwatts at 20 mK, in principle it is feasible to integrate up to millions of the combs at the base temperature, based on their energy efficiency. This presents a unique advantage over semiconductor-based combs.

Comment#11:

11. In the last paragraph, the authors brought a very interesting application of their comb, which is on-chip arbitrary waveform generation. I totally agree with this point, but would like to know if this method has specific advantages over cryogenic CMOS-based waveform generation.

Our response

Due to the relatively large dissipation of CMOS-based waveform generators, integrating them directly at the millikelvin temperature range, required for quantum circuits, presents a significant challenge. The cryogenic CMOS-based waveform generators typically operate at 1 to 4 K. In contrast, our energy-efficient comb could offer a promising technique for arbitrary waveform generation capable of operating directly at millikelvin temperatures. Of course, substantial future developments and engineering efforts are necessary.

Responses to comments and recommendations of Reviewer 2

Reviewer recommendation:

This transformational technique of generating frequency combs with a very low current injection and coupling to a superconducting resonator is inspired and high quality work. The addition of the injection locking is very important, but the observation of the coherent subcombs is the most useful in terms of confirming that this is a coherent comb structure.

Our response

We appreciate your positive feedback on our work. Your insights are precious to us and help us to improve the quality of our research.

Comment#1:

If there is any negatives, it seems that there are statements about the possibility of developing on-chip AWGs go too far and may actually distract from the achievements outlined. For example, the implementation of that involves pulse-shaping methods that may or may not be available at cryogenic stages.

Our response

We apologize for any confusion caused by our prospective statements regarding AWGs. Frequency combs have a well-recognized application in the development of Arbitrary Waveform Generators (AWGs). Reference 52 outlines how pulse or waveform shaping can be accomplished by adjusting the intensities of the combs teeth. In the context of on-chip AWGs based on the superconducting comb, one possible approach to control the intensity of each comb tooth could involve utilizing a series of frequency-tunable superconducting resonators. The tunable resonators could be designed to align with each comb mode, serving as tunable microwave filters to modulate the output of each comb tooth. Importantly, the tunable resonators are compatible with integration at cryogenic temperatures, as evidenced by the work presented in Phys. Rev. Appl., 19, 034021 (2023). We have expanded our manuscript to include a statement and references regarding this possible method for the development of on-chip AWGs in the section of ‘Discussion and Prospects’.

Comment#2:

While it can easily be calculated what power is in the peaks, it would be useful for the authors to estimate that and put in the context of quantum circuits, i.e. drives and readouts, so the applicability for these generators at base temperature can be easily translated to that technology.

Our response

Thank you for this insightful suggestion. We have calculated the power of each comb peak, and the results are presented in Fig. S5 below (as Supplementary Figure 13 in the

revised manuscript). The power of each peak ranges from 0.65 fW to 0.306 pW, consistent to the output power of the single-mode source reported in Ref. 25. Given that the comb emission bandwidth is significantly broader than our detection bandwidth, estimating the total emission power presents a challenge. Moreover, the output power of Josephson junction coupled superconducting resonators exhibits substantial variation across devices [refer to Refs. 25 and 26], suggesting that future enhancements in the comb's output power is possible through optimizing the device's structures and/or parameters [26]. Ref. 26 shows that the Josephson junction-based microwave source is compatible for controlling quantum circuit in terms of power and noise. Finally, the on-chip input power of our comb, as estimated from the I-V characteristics, spans from 40 pW to 600 pW. This is considerably lower than the cooling power of hundreds of microwatts available in a typical commercial dilution refrigerator, thereby affirming compatibility with base temperature integration for quantum circuit operation. We have added the data to the Supplementary Information as Supplementary Figure 13 and added a corresponding discussion in the revised manuscript.

Recommendation:

Overall an excellent idea and well organized and written paper.

Our response

Thank you once again for your generous feedback. Your kind words are a great source of motivation for us, and your insightful comments have been invaluable in enhancing the quantity of our work.

REVIEWERS' COMMENTS

Reviewer #1 (Remarks to the Author):

I first would like to thank the authors for their thoughtful comments to my suggestions and questions. However, the revised manuscript seems not properly reflect their comments written in the response letters.

1. For example, I asked them if the authors could provide more clear motivation with more emphasis on microwave frequency combs. But the introduction part did not reflect this comment. I don't think it is proper to compare their microwave frequency combs with the drawbacks of optical combs as their operating frequencies and applications are totally different.

Therefore, the statements

"However, most on-chip frequency combs, such as integrated semiconductor mode-locked lasers and microresonator-based Kerr combs, require sophisticated engineering to hybridize multiple materials into complex structures, demanding substantial manufacturing efforts [5]. Additionally, many combs involve intricate laser tuning and control procedures for operation, necessitating challenging integration of expensive optical and electronic components [16]. Another crucial limitation for large-scale integration of semiconductor-based combs is their relatively high energy consumption, stemming from high pumping thresholds, preventing their use in energy-sensitive applications."

in the first paragraph is not convincing for their current work, meaning that the sentences should not be in front of

"Here, we address these challenges by introducing an all-superconductor-based microcomb, featuring an elegantly simple structure, effortless operation, and extremely low power consumption."

because the former is not the issue for superconducting combs!

2. Moreover, the authors provided comprehensive explanation with additional figures in their response letter. However, it seems like they just put them in the supporting information without mentioning in the revised manuscript. They should provide some key statements from their responses in the revised manuscript and try to relate these statements with corresponding supporting figures at least.

Again, I am quite satisfied with the responses from the authors, but before its publication, comments from the reviewers (and the author's responses) should be well reflected in the revised manuscript.

Reviewer #2 (Remarks to the Author):

I support publication of the revised manuscript as-is.

Responses to comments of Reviewer 1

I first would like to thank the authors for their thoughtful comments to my suggestions and questions. However, the revised manuscript seems not properly reflect their comments written in the response letters.

Our response

Thank you once again for your thoughtful comments and suggestions. We have carefully revised the manuscript to ensure it fully reflects the detailed points addressed in our previous response letters. Your feedback continues to be invaluable in enhancing the quality of our work.

1. For example, I asked them if the authors could provide more clear motivation with more emphasis on microwave frequency combs. But the introduction part did not reflect this comment. I don't think it is proper to compare their microwave frequency combs with the drawbacks of optical combs as their operating frequencies and applications are totally different.

Therefore, the statements

“However, most on-chip frequency combs, such as integrated semiconductor mode-locked lasers and microresonator-based Kerr combs, require sophisticated engineering to hybridize multiple materials into complex structures, demanding substantial manufacturing efforts [5]. Additionally, many combs involve intricate laser tuning and control procedures for operation, necessitating challenging integration of expensive optical and electronic components [16]. Another crucial limitation for large-scale integration of semiconductor-based combs is their relatively high energy consumption, stemming from high pumping thresholds, preventing their use in energy-sensitive applications.”

in the first paragraph is not convincing for their current work, meaning that the sentences should not be in front of

“Here, we address these challenges by introducing an all-superconductor-based microcomb, featuring an elegantly simple structure, effortless operation, and extremely low power consumption.”

because the former is not the issue for superconducting combs!

Our response

We have removed the discussion of the drawbacks of optical combs in the first paragraph and have included the following discussion to emphasize the significance of microwave frequency combs.

“However, most on-chip frequency combs, such as integrated semiconductor mode-locked lasers and microresonator-based Kerr combs, mainly operate in the optical frequency domain. A fully integrated frequency comb functioning in the microwave domain remains elusive, impeding the advancement of chip-based microwave spectroscopy and integrated quantum circuits, which typically require precise microwave control.”

2. Moreover, the authors provided comprehensive explanation with additional figures in their response letter. However, it seems like they just put them in the supporting information without mentioning in the revised manuscript. They should provide some key statements from their responses in the revised manuscript and try to relate these statements with corresponding supporting figures at least.

Our response

We have added a new paragraph (detailed below) to provide the comprehensive information on the additional figures, which cover the comb’s stability, linewidth, and injection power dependent results. All supplementary figures are now thoroughly discussed and cited in the revised manuscript.

“The emission power of each individual comb line, ranging up to subpicowatt (see Supplementary Figure 3), is consistent with that of the continuous-wave (or single-mode) source [25]. The microcomb's performance could be further enhanced by refining the structures and parameters of the Josephson junction-coupled resonators [26]. The stability of the frequency comb is determined by the linewidth of the spectral lines (see Supplementary Figure 4). In general, the overall linewidth is influenced by the quality factor of the resonator, the stability of the DC bias voltage, and environmental electromagnetic and thermal noise. Moreover, our experiments demonstrate that the linewidth of our comb increases quadratically with the mode number (or frequency), as illustrated in Supplementary Figure 5. This finding is consistent with the observations in quantum-limited optical combs [35, 36].”

Again, I am quite satisfied with the responses from the authors, but before its publication, comments from the reviewers (and the author’s responses) should be well reflected in the revised manuscript.

Our response

We sincerely appreciate your insightful comments and suggestions. We have incorporated all the significant responses and corresponding revisions into the main text of the revised manuscript.